# Silicon Nanomaterials Enhance Seedling Growth and Plant Adaptation to Acidic Soil by Promoting Photosynthesis and Antioxidant Activity in Mustard (*Brassica campestris* L.)

**DOI:** 10.3390/ijms251910318

**Published:** 2024-09-25

**Authors:** Md. Kamrul Hasan, Jannat Shopan, Israt Jahan, Tonima Islam Suravi

**Affiliations:** 1Department of Agricultural Chemistry, Sylhet Agricultural University, Sylhet 3100, Bangladesh; isratbsau2015@gmail.com (I.J.); tonimaislam5205@gmail.com (T.I.S.); 2Key Laboratory of Integrated Research in Food and Agriculture (IRFA), Sylhet Agricultural University, Sylhet 3100, Bangladesh; jannat.shopan.197@hau.ac.bd; 3Department of Haor and Hill Agriculture, Habiganj Agricultural University, Habiganj 3300, Bangladesh

**Keywords:** soil acidity, silicon nanomaterials, redox homeostasis, yield, farming resilience

## Abstract

Soil acidity is a divesting factor that restricts crop growth and productivity. Conversely, silicon nanomaterials (Si-NMs) have been praised as a blessing of modern agricultural intensification by overcoming the ecological barrier. Here, we performed a sequential study from seed germination to the yield performance of mustard (*Brassica campestris*) crops under acid-stressed conditions. The results showed that Si-NMs significantly improved seed germination and seedling growth under acid stress situations. These might be associated with increased antioxidant activity and the preserve ratio of GSH/GSSG and AsA/DHA, which is restricted by soil acidity. Moreover, Si-NMs in field regimes significantly diminished the acid-stress-induced growth inhibitions, as evidenced by increased net photosynthesis and biomass accumulations. Again, Si-NMs triggered all the critical metrics of crop productivity, including the seed oil content. Additionally, Si-NMs, upon dolomite supplementation, further triggered all the metrics of yields related to farming resilience. Therefore, the present study highlighted the crucial roles of Si-NMs in sustainable agricultural expansion and cropping intensification, especially in areas affected by soil acidity.

## 1. Introduction

Low soil pH is an important factor in soil chemistry that largely influences soil nutrient cycling by affecting the soil microbial community and diminishing soil health and fertility. The unavailability of essential nutrients, like calcium, magnesium, phosphorus, sodium and potassium, by means of leaching or fixation leads to deficiency, poor plant growth and low crop productivity [1,2]. The key interconnected phenomenon of low soil pH-induced nutrient deficiency is the generation of excess reactive oxygen species (ROS) in plants that impact plant heath and productivity. Usually, excess ROS accumulation in plants oxidizes the vital cell ultrastructure organelles and inactivates the general cellular functions; as a result, plants not only become vulnerable to adaptation in acidic soils but also show reduced productivity. For example, it has been observed that the grain filling in crops is strongly affected by low pH through contributing to enhanced lipid peroxidation or membrane permeability [3]. However, the possessed antioxidant system may be activated to quench the excess ROS in plants during stress. Shockingly, such an activation of the defense system may not be sufficient to protect the oxidative damage to the plant over the long term [3,4]. Moreover, low soil pH could impact the effectiveness of plant antioxidant defense systems indirectly through nutrient misbalancing in cellular biosynthetic pathways [5].

Acid soils occupy approximately 30–40% of the world’s land surfaces and about 50% of arable land, which affects germination niche, plant adaptation and the overall crop growth and productivity [6,7]. It has been observed that acid soil causes a burden of about USD 6.0 billion, which is 6% of the value of the total current production [8]. An extensive number of studies have explored how acid stress affects crop productivity via the inhibition of root elongation and branching and the reduced membrane permeability of roots for water and nutrient uptake [7]. Moreover, low soil pH augmented the replacement of essential base cations with toxic elements like aluminum (Al) by promoting their solubility, creating widespread toxicity. It has been observed that high levels of aluminum- or manganese-induced redox imbalances are the key vulnerability behind damage to plant root systems in acidic soils that reduce not only the water and nutrient absorption but also lead to stunted growth and productivity [7,9]. Therefore, understanding the interplay between soil acidity and ROS homeostasis is essential for developing strategies to improve plant health and productivity in acidic soils.

In contrast, there is an increasing demand for the expansion of agriculture to unfavorable soils, like those of low soil pH, to meet the global demand for food security [10]. Hence, sustainable management practices are crucial to maintain acid soils and stimulate agricultural productivity. Importantly, the efforts to increase and/or sustain the agricultural productivity are often driven by the innovation of technologies and effective management practices that could mitigate the adverse effects of soil acidity and oxidative stress. Expectedly, in recent years, the application of biostimulants like hormones or engineered nanoparticles resembling silicon nanomaterials (Si-NMs) has been widely used in agriculture as an eco-friendly approach to enhance the crop productivity and tolerance to multiple stresses through the addition of limestone [11]. Primarily, silicon is taken up by plants as monosilicic acid (H_4_SiO_4_) through specific transporters and deposited in tissues as amorphous silica (SiO_2_•nH_2_O), which regulates plant growth under adverse conditions like soil acidity [12]. However, as a complex abiotic factor, the overwhelming role of low soil pH or soil acidity on plant survival potential has been poorly explored, and very few studies have focused on the involvement of Si-NMs in plant adaptation to harsh acidic soil [13]. Therefore, we hypothesized that the application of Si-NMs might provide an additional strength to reduce the vulnerability of plants to soil acidity. Our study provides a comprehensive understanding of Si-NM-mediated effects on seed germination, seedling growth and their association with antioxidant activity and redox homeostasis. Moreover, we also studied the influential role of Si-NMs with or without dolomite as a global soil correction practice on photosynthesis, yield attributes and seed yield of mustard with an oil percentage essential for sustainable agricultural development and to address the global food demand.

## 2. Results

### 2.1. Effect of Differential Doses of Si-Nanomaterials (Si-NMs) on Seed Germination and Seedling Growth of Mustard under Acidic Conditions

To understand the heightened effect of Si nanomaterials on mustard germination and seedling growth, we first performed an in vitro experiment with different doses (0, 0.25, 0.5, 1.0 and 1.5 mmol L^−1^) of Si-NMs and observed the competency at 7 days after sowing. The results showed that all the differential doses of Si nanoparticles had stimulatory effects on seed germination and seedling growth under normal conditions, as visualized in Figure 1. However, among the doses, 0.5 mmol L^−1^ showed the finest results, followed by others. Like the visual display, our numerical data also revealed that the 0.5 mmol L^−1^ dose of nano-Si enhanced, by about 8.7% and 7.8%, additional germination in BARI mustard-14 and BARI mustard-17 compared to their respective control. Similarly, the seedling height and biomass accumulation were also augmented by seed pretreatment with Nano-Si. For example, the results showed that, among the doses, 0.5 mmol L^−1^ Nano-Si enhanced the root and shoot height by about 140% and 54.8% in BARI mustard-14 and 189.3% and 47.2% BARI mustard-17, respectively, compared to their control, which was followed by 1.0 mmol L^−1^ (Figure 1d,e). In contrast, higher doses (1.5 mmol L^−1^) of Nano-Si displayed 15.9% and 14.5% reduced biomass accumulation and 6.2% and 11.2% reduced germination in BARI mustard-14 and BARI mustard-17, respectively, compared to 0.5 mmol L^−1^ doses, indicating that Si-NM-mediated mustard seed germination and seedling growth are dose-dependent.

After preliminary selection of the Nano-Si dose (0.5 mmol L^−1^), we performed another experiment under acidic conditions at pH 4.5 to observe the efficacy of Nano-Si in stressed conditions (Figure 2), as soil acidity is one of the major concerns that strongly reduces mustard seed germination, seedling growth and adaptation. Interestingly, the results showed that Nano-Si significantly improved the germination and seedling biomass accumulation in both BARI mustard-14 and BARI mustard-17, which were firmly suppressed by the acidic medium. The results showed that under acidic conditions, the germination and biomass accumulation of BARI mustard-14 and BARI mustard-17 was reduced by 55.3% and 61.8% and 21.7% and 31.8%, respectively, compared to their control, indicating that BARI-14 is more sensitive to acid stress compared to BARI-17 (Figure 2a–e). However, seed pretreatment with Nano-Si improved germination and biomass accumulation in both varieties. For example, in BARI-14, the germination and biomass accumulation were improved by 1.4-fold and 3.4-fold, respectively, compared to their acid-only treatment, similar to the response of BARI-17. These results indicated that Nano-Si has a positive stimulatory effect on mustard seed germination, seedling growth and biomass accumulation under stressed acidic conditions, while this response varied with the varietal consistencies.

### 2.2. Si-NM-Mediated Mustard Seedling Growth and Establishment Are Associated with ROS Homeostasis

To understand the interaction between acidic growth conditions and the generation of ROS, we stained the mustard seedlings with DAB and NBT to visualize the H_2_O_2_ and O_2_*^−^. The histochemical results showed that ROS accumulation remained almost constant in the control condition, while ROS were substantially accumulated in acidic conditions (Figure 3a,b). However, seed pretreatment with Si-NMs strongly minimized the acidic-growth-condition-induced ROS accumulation in both mustard varieties. Similar to the visualized symptoms, our biochemical values also revealed that the acidic growth condition augmented a more than 2.0-fold higher H_2_O_2_ accumulation in both BARI mustard-14 and BARI mustard-17, compared to their respective control. However, seed pretreatment with Si-NMs reduced about 22.3% and 30.7% less H_2_O_2_ accumulation in both BARI mustard-14 and BARI mustard-17 seedlings compared to their acid-only treatment (Figure 3c). Furthermore, the malondialdehyde (MDA) content from lipid peroxidation and the percent of electrolyte (%EL) were also found to be increased in both mustard seedlings under acidic growth conditions, while seed pretreatment with Si-NMs minimized the excessive MDA and electrolyte leakage content. For example, seed pretreatment with Si-NMs reduced the EL by about 38.6–45.8% MDA and 30.8–36.9% in both tested varieties (Figure 3d,e), indicating the potential role of Si-NMs in ROS homeostasis.

To further investigate the physiological interplay of Si-NMs on redox homeostasis under acidic conditions, we measured the total antioxidant, DPPH and the ratio of GSH/ GSSG and AsA/DHA. Interestingly, the results displayed that the total antioxidant and DPPH activity was strongly inhibited by acidic growth conditions in both varieties. For example, the total antioxidant and DPPH activity was reduced by 30.7% and 8.9% and 22.8% and 19.9%, respectively, in BARI mustard-14 and BARI mustard-17 compared to their control conditions (Figure 4a,b). In contrast, although seed pretreatment with Si-NMs under normal conditions showed no effect, Si-NMs significantly hindered the reduced activity of DPPH and the total antioxidant content under acidic growth conditions, showing about 47.8% and 65.4% and 91.8% and 96.0% higher activity in BARI mustard-14 and BARI mustard-17 compared to their acid control (Figure 4). Similarly, the ratio of reduced glutathione to oxidized forms of glutathione (GSH/GSSG) and the ratio of ascorbate to dehydroascorbate (AsA/DHA) in both mustard seedling varieties were strongly reduced under acidic growth conditions compared to their respective control (Figure 4c,d). The reduction in the GSH/GSSG and AsA/DHA ratios ranges from 73.4 to 103.7% and 84.1 to 112.3%, respectively, under acidic conditions compared to their control conditions. In contrast, although the seed pretreatment with Si-NMs revealed no significant change in the ratio of GSH/GSSG and AsA/DHA under normal growth conditions, but under acidic growth conditions, Si-NMs prominently uphold the ratio of GSH/GSSG and AsA/DHA in the seedlings of both tested mustard varieties (Figure 4c,d). For example, the ratios of GSH/GSSG and AsA/DHA in seedlings of BARI mustard-14 were increased by about 68.5% and 76.8%, respectively, under acidic growth conditions compared to non-treated seedlings. These results indicate that seed pretreatment with Si-NMs plays a critical role in redox homeostasis under acidic conditions by stimulating the antioxidant and DPPH activity and upholding the ratio of GSH/GSSG and AsA/DHA.

### 2.3. Exogenous Si-NMs Enhanced Plant Growth and Biomass Accumulation under Acidic Field Soil Condition

Soil acidity is a limiting factor of crop growth and productivity. To understand the promising opportunities of Si-NMs to manage the adverse soil acidity effect and improve crop productivity, we extended our study at the field level under acidic soil conditions. The results displayed that soil acidity (pH 4.5–4.8) in field conditions strongly suppressed the normal growth of both mustard varieties compared to the crops grown at the normal pH 6.2–6.3 (field soil) level (Figure 5a). However, the exogenous application of Si-NMs at the selected dose (0.5 mmol L^−1^) significantly improved the acidic-soil-condition-induced suppression of mustard crop growth. For example, the total biomass accumulation of BARI mustard-14 and BARI mustard-17 was reduced by 4.1-fold and 1.2-fold, respectively, compared to the plants grown in normal acidic soil conditions (Figure 5d). However, the exogenous application of Si-NMs improved the total biomass accumulation by about 3.7-fold and 1.9-fold compared to the non-treated plants grown in acidic soil conditions. Similarly, the total chlorophyll content, net photosynthesis and root shoot ratio of acid-stressed plants were also improved via the exogenous application of Si-NMs in both tested varieties. For instance, the total chlorophyll content and net photosynthesis were improved by 35.1% and 76.4% and 157.2% and 200.4% for BARI mustard-14 and BARI mustard-17, respectively, compared to the acid-stressed plants. In contrast, the exogenous application of Si-NMs caused a further increase in the total chlorophyll content, net photosynthesis and growth biomass in both mustard varieties grown in normal field soil conditions. The results showed that in normal field soil conditions, Si-NMs stimulated further plant biomass accumulation by about 48.9% and 80.7% in BARI mustard-14 and BARI mustard-17, respectively, compared to their respective control (Figure 5d).

Although the crop growth significantly improved upon Si-NM application under both the acidic and normal field conditions, under the acid-stressed condition, the total chlorophyll content, net photosynthesis and biomass accumulation growth contribution remained behind the optimized level of crops cultivated in normal field soil conditions. Hence, as a soil management strategy, we also applied the dolomite at a rate of one ton ha^−1^ and observed the potential of Si-NMs on the growth attributes of mustard crops. Interestingly, the results showed that only dolomite supplementation failed to diminish the acid-stress-induced delay in mustard crop growth and biomass accumulation (Figure 5). For example, the dolomite-only treatment showed 20.2% and 17.7% and 18.9% and 7.2% reduced net photosynthesis and biomass accumulation, which was statistically insignificant, with the Si-NM-only treatment in BARI mustard-14 and BARI mustard-17, respectively, compared to their respective field soil control (Figure 5b,d).These results indicate that neither dolomite nor Si-NMs are independently capable of optimizing the acid-stress-induced growth suppression of mustard crops. However, the application of Si-NMs upon dolomite supplementation as a soil amendment completely reversed the acid-stressed-induced suppression of mustard crop growth, as evident by the chlorophyll content, net photosynthesis and biomass accumulation in both tested varieties (Figure 5). For example, regarding the application of Si-NMs upon dolomite supplementation, the net photosynthesis and biomass accumulation increased about 3.9% and 10.5% and 15.8% and 39.7% in BARI mustard-14 and BARI mustard-17, respectively, compared to the crops grown in normal field conditions.

### 2.4. Effects of Si-NMs on Yield Attributes of Mustard Crop under Acidic Field Soil Condition

Yield attributes are critical metrics responsible for crop productivity. To observe the efficiency of Si-NMs on the yield attributes of mustard, we recorded the total numbers of siliqua per plants, number of seeds per siliqua and hundred seeds weight under both acid-stressed and field soil conditions, with or without Si-NMs and/or dolomite. The results displayed that siliqua size largely varied among the treatments; specifically, the tiniest sized siliqua was observed in both tested varieties under acid soil conditions (Figure 6a). However, the siliqua sizes were significantly enlarged upon the independent application of both the Si-NMs and dolomite as a soil supplementation. Conversely, their combined application caused a further increase in the siliqua and even the leaf size under acid soil conditions. In accordance with the physiological display, the numerical values also showed that the total number of siliqua plants^−1^ significantly increased in both BARI mustard-14 and BARI mustard-17 under acid soil conditions upon Si-NM or dolomite application. The results showed that, compared to the field soil control, the total number of siliqua plants^−1^ reduced by 173.9% and 90.1% in BARI mustard-14 and BARI mustard-17 under acid soil conditions, showing only 23.4% reduced siliqua in BARI mustard-14 but 84.1% increased siliqua in BARI mustard-17 upon Si-NM application, indicating varietal differences of adaptive response. On the other hand, only dolomite application as a soil supplementation had a much lower effect and showed about 52.1% and 24.8% reduced siliqua settings plant^−1^ under acid soil compared to field soil control. However, their combined application showed about 71.5% and 161.4% increased numbers of siliqua in BARI mustard-14 and BARI mustard-17 under acid soil conditions, which were statistically insignificant or little tuned with Si-NM treatment in field soil (FS) conditions (Figure 6b). Similarly, the total number of seeds per siliqua and the hundred seeds weight were also reduced by 14.8% and 50.1% and 38.1% and 48.6%, respectively, in BARI mustard-14 and BARI mustard-17 under acid soil conditions compared to the plants grown in field soil with a normal pH. The application of exogenous Si-NMs significantly improved the traits but failed to optimize; however, it caused a further increase in the total number of seeds per siliqua and the hundred seeds weight in plants grown in field soil conditions. In contrast, the counting remained statistically similar or showed a slight change with the counting in plant soils treated with dolomites (Figure 6c,d). However, their combined application completely reversed the acid-stress-induced suppression of total seed counting per siliqua and the hundred seeds weight. These results indicated that the combined application of Si-NMs and dolomite, instead of their solo use, plays a potential role in promoting the yield attributes of mustard crops suppressed by acid soil conditions.

### 2.5. Si-NMs Enhanced Seed Yield and Oil Percentage of Mustard Reduced by Acidic Field Soil Condition

To understand the role of Si-NMs on the sustainable intensification of mustard crops under stressful conditions, we examined the mustard seed yield and analyzed the oil content for economic resilience. The results showed that the mustard seed yield and oil content in seeds were highly susceptible to low soil pH and showed 2.9-fold and 2.7-fold reduced seed yield and 25.7% and 22.2% reduced oil content in BARI mustard-14 and BARI mustard-17 under acid soil conditions compared to the plants grown in normal field soil conditions (Figure 7). However, the exogenous supplementation of Si-NMs significantly improved both the seed yield and oil content in both tested varieties. Upon Si-NM application, the seed yield increased by about 155.3% and 126.5% and the oil content increased by about 16.5% and 10.3% in BARI mustard-14 and BARI mustard-17, respectively, in acid soil conditions. Interestingly, these results were mostly statistically insignificant with the soils cultivated upon dolomite supplementation as a soil amendment and completely failed to optimize the acid-stress-induced suppression. However, their combined application caused a further increase in both the seed yield and oil content and remained statistically insignificant with the field soil controls (Figure 7). In contrast, the exogenous supplementation of Si-NMs to the crops cultivated at normal soil pH showed further augmentation in the seed yield and oil content in both tested varieties. Compared to the field soil control, the seed yield and the oil content were increased by about 24.1% and 12.8% and 6.2% and 8.1%, respectively, in BARI mustard-14 and BARI mustard-17 in Si-NM-treated field soil crops, indicating the economic resilience of Si-NMs, essential for sustainable agricultural developments.

## 3. Discussion

The growing concern for global food security has expanded to the further expansion and intensification of agriculture, particularly the region of soils affected by acidity [10]. However, soil acidity or low soil pH is thought to be one of the key stress factors restricting the total acreage and crop productivity, subsequently threatening global food security [14]. However, the most recent year’s nano-fertilizers like silicon nanomaterial (Si-NMs) raised a healthier option for sustainable agricultural development and is observed to improve growth, physiological development and biochemical response in plants under diverse stress conditions, like drought, salinity or metal stress [15,16]. Regarding the benefits of Si-NMs as an ecologically sustainable and economically stable food production strategy, we designed the present experiments to understand its suitability for mustard crop production under acidic soil conditions, which remains elusive. To understand the influential effect of Si-NMs, we performed a sequential experiment, from seed germination to field trial, of mustard crops. The results showed that, among the deferential doses, 0.5 mmol L^−1^ of Si-NMs showed the finest stimulatory effects on seed germination and seedling growth that was firmly suppressed by the acidic medium (Figure 1 and Figure 2). The promotion of growth associated with the management of the adverse effects of acid stress dominantly minimized the generated ROS, lipid peroxidation (MDA) and electrolyte leakage (EL) by stimulating the total antioxidant activity, DPPH and the ratios of GSH/GSSG and AsA/DHA (Figure 4). Furthermore, our experimental results also showed that Si-NMs are promising to manage the adverse effects of soil acidity and improve crop productivity at the field level, as evident by the improved chlorophyll content, net photosynthesis, biomass accumulation and the metrics of yield attributes upon Si-NM application in both tested varieties, BARI mustard-14 and BARI mustard-17 (Figure 5, Figure 6 and Figure 7). However, the application of Si-NMs upon dolomite supplementation as a soil amendment completely reversed the acid-stress-induced suppression of mustard crop growth, seed yield and oil content, indicating the economic resilience of Si-NMs, essential for the sustainable intensification of agricultural growth [17,18].

Germination is a critical aspect of the lifecycle and plays a vital role in successive regeneration, ecological balance and food production [19]. To identify the threshold-type response of Si-NMs to low pH, initially, we observed the seed germination competency with different doses (0, 0.5. 1.0 and 1.5 mmol L^−1^) of Si-NMs in BARI mustard-14 and BARI mustard-17 (Figure 1). The experimental results showed that Si-NMs mediated mustard seed germination and seedling growth extremely dose-dependently, and 0.5 mmol L^−1^ showed the finest results. However, higher doses (1.5 mmol L^−1^) of Nano-Si displayed 6.2% and 11.2% reduced rates of germination and 15.9% and 14.5% decreased biomass accumulation in BARI mustard-14 and BARI mustard-17, respectively, compared to 0.5 mmol L^−1^ doses. The toxicity of nanoparticles in plants at higher doses is limited but thought to be regulated by a complex phenomenon of multiple intrinsic factors [20]. One common hypothesis by scientists suggests that at higher doses, Si-NMs might produce excess metal ions in plant tissues that promote the generation of reactive oxygen species (ROS) in cells, leading to cellular structure and macromolecular damage [21,22]. Moreover, Si-NMs at higher doses might interfere with the hormonal balances that restrict the overall germination process. For example, it has been reported that Ag-NPs alter the hormonal balance at higher doses in many species, including Arabidopsis, pepper, cucumber and wheat and shakes the germination and growth [23]. Therefore, we selected the finest dose (0.5 mmol L^−1^) of Si-NMs for further experimentation under acidic or low pH 4.5 conditions, a major limiting factor restricting mustard seed germination, seedling growth and adaptation [24,25]. Interestingly, the experimental findings showed that Si-NMs dramatically upgraded the germination, seedling growth and biomass accumulation in both tested varieties that were firmly suppressed by the acidic medium (Figure 2); however, these positive stimulatory responses change from variety to variety [26]. Though the mechanism remains unclear, Si-NM-mediated enhanced mustard seed germination under stressed conditions may be associated with increased water absorption through enhanced aquaporin gene expression that stimulates the metabolic activity in seed results, thus improving germination [20]. As optimum water or moisture content in the seed is the key factor of metabolic degradation carbohydrates to sugar and the increased activity of amylase and protease allow for quicker cell division in the embryo, this consequently enhanced the normal germination that was halted at <0.1 g H_2_O content in per gram seeds weight [20,27]. Previously, it has also been observed that seed priming with Si-NMs enhanced not only the germination but also improved the seedling growth and biomass accumulation by improving stress tolerance, similar to our current experimental findings [28].

Generally, the active mitochondria in germinating seeds are likely one of the principal sites of ROS generation that initially produce superoxide anions (O_2_^•−^) and then hydrogen peroxide (H_2_O_2_). However, under normal growth conditions, the generated ROS were scavenged by the antioxidant enzymatic activity, such as SOD, POD, CAT, APX, AsA and GSH [29]. In accordance, our present experimental findings also displayed that under acid stress conditions, mustard seedlings showed higher O_2_^•−^ and accumulation of H_2_O_2,_ resulting in increased malondialdehyde (MDA) content and the percent of electrolyte leakage in both tested varieties (Figure 4). These might be due to the rate-limiting antioxidant potential of seedlings [4,30]. However, seed priming with Si-NMs showed an enhanced activity of total antioxidant activity, DPPH and the ratios of GSH/GSSG and AsA/DHA in both mustard varieties, BARI mustard-14 and BARI mustard-17, under acidic conditions, indicating the critical interplaying role of Si-NMs in improving redox homeostasis (Figure 3 and Figure 4) [31]. Accordingly, a number of studies recently explored that, as an innovative technique, Si-NMs could be an ecologically sound alternative to mitigate the oxidative stress generated in plants under stressed conditions [32]. Si-NMs improve the DPPH and total antioxidant activity up to 65.4% and 96.0% in mustard seedlings under acid stress conditions compared to their respective control, in addition to upholding the ratios of GSH/GSSG and AsA/DHA (Figure 4). Moreover, very recently, Si-NMs were found to control the LOX and redox-related gene expression that decreased ROS-induced cellular damage under stressful conditions, consequently enhancing the plant adaptation potential to adverse conditions like soil acidity [29].

As a critical factor, low soil pH not only regulated the seed germination but also negatively impacted the plant adaptation and crop productivity results, affecting the liveli-hoods of farmers across the world [1]. Considering soil acidity as the most divesting factor, we also extended our experiments at the field level, specifically in soil consisting of low pH (4.5–4.8), and evaluated the opportunities of Si-NMs for managing the adverse effects of soil acidity. Increased H^+^ ions in soil are a critical indicator of the evaluation of environmental pollution [6]. Our field experimental results showed that soil acidity critically regulated the normal growth of both tested varieties, as evident by the reduced chlorophyll content, net photosynthesis and root shoot ratio and biomass accumulation (Figure 5). Many recent studies unveiled that soil acidity affects every aspect of plant growth and productivity by inhibiting nutrient availability, root development and soil microbial activity [13,33,34]. Interestingly, our results showed that the exogenous application of Si-NMs improved net photosynthesis by about 157.2% and 200.4% and the total biomass accumulation by about 3.7-fold and 1.9-fold in BARI mustard-14 and BARI mustard-17, respectively, compared to the non-treated plants, flattening their use for sustainable agricultural expansion to the acid stress region [35].

Again, the yield attributes, such as the total numbers of siliqua per plants, number of seeds per siliqua and hundred seeds weight of mustard, are thought to be the critical metrics responsible for crop productivity [36]. Surprisingly, the results displayed that all the yield attribute metrics significantly improved upon feeding with Si-NMs in both tested varieties, thereby increasing the total seed yield that was strongly suppressed by soil acidity (Figure 6 and Figure 7). For example, upon Si-NM application, the seed yield increased by about 155.3% and 126.5% in BARI mustard-14 and BARI mustard-17, respectively, in acid soil conditions (Figure 7a). In accordance, a number of studies explored that Si-NMs enhanced crop growth and productivity in numerous crop species, such as cucumber, pea, wheat, oat, maize, grape, bean, rice, etc., under diverse types of stress, like salinity, water deficiency or heavy metal stress [16,37]. Generally, Si is primarily taken up by plants as monosilicic acid (H_4_SiO_4_) through specific transporters in roots and then various tissues, where they are deposited as amorphous silica (SiO_2_•nH_2_O), regulating the plant growth and adaptation to adverse conditions like soil acidity [12]. This means that the stimulatory potential of Si-NMs might be associated with enhanced cellular protection by promoting antioxidant activity and redox homeostasis. Moreover, Si-NMs have been found to improve lateral root or even main root growth, which eventually increases the nutrient uptake potential of plants, leading to crop yields [38]. Despite the mustard seed yield Si-NMs being found to increase the oil content in seeds (Figure 7a), it has been observed that under low soil pH conditions, the oil content in BARI mustard-14 and BARI mustard-17 increased by about 16.5% and 10.3%, respectively, compared to non-treated plants, signifying the economic resilience of farmers [1].

However, Si-NM application to optimize the crop growth and yield performed poorly in extremely low soil pH conditions [39]. Similarly, we also found that Si-NMs failed to completely compensate the yield and yield attributes of mustard crops when compared with crops grown in field soil (slightly acidic pH 6.2–6.3) conditions (Figure 5, Figure 6 and Figure 7). In contrast, in field soil or slightly acidic conditions, Si-NM application caused further augmentation of all the metrics associated with farming resilience. Hence, as a soil management strategy, we also applied the dolomite at a rate of one ton ha^−1^ as a global acid soil correction practice, which consequently reduced the crop sensitivity to low soil pH [33]. For example, Si-NM application upon dolomite supplementation increased the seed yield and the oil content by about 24.1% and 12.8% and 6.2% and 8.1%, respectively, in BARI mustard-14 and BARI mustard-17 compared to the crops grown in field soil conditions, indicating the wider feasibility of Si-NM application across diverse soil pH ranges [39].

## 4. Materials and Methods

### 4.1. Plant Materials and Seed Pretreatment with Silicon Nanomaterials (Si-NMs)

Seeds of two mustard (*Brassica campestris*) varieties, BRAI Mustard-14 and BRAI Mustard-17, were collected from Bangladesh Agricultural Research Institute (BARI), Gazipur, Dhaka, Bangladesh. After assortment, healthy seeds were sterilized with 10% sodium hypochloride solution for five minutes and then repeatedly rinsed with distilled water. Once washed, the seeds were soaked with graded level of silicon nanomaterials (Si-NMs) at rate of 0, 0.25, 0.5, 1.0 and 1.5 mmol L^−1^ for about twelve hours to observe the competency. The Si-NMs with average primary particle sizes 30 nm and 99.9% pure at metal basis were purchased from the Aladdin Industries Corporation, Shanghai, China. The measured amount of Si-NMs with distilled water was sonicated for thirty minutes to prepare the homogenous mixture using sonic’s vibra-cell (Model-VCX 505, Gütersloh, Germany), as previously discussed [28].

After seed pretreatment with differential doses of Si-NMs, three hundred seeds of each treatment were transferred into three Petri dishes (60 mm × 15 mm), comprising a double layer of filter paper. The filter papers were soaked with distilled water to ensure seed moisture during germination. In case of acid treatment, the pretreated seeds with or without Si-NMs placed into the Petri plate containing acidified water (pH-4.5) soaked the double-layered filter paper. The acidified water prepared by adding the HNO_3_ drop by drop to the tap water and adjusted at 4.5 pH level using a benchtop pH meter (Model-pH 2601, Ningbo, China), as previously described [40]. All the in vitro experiments were performed in a growth chamber (model-ICB-L250B, Bioevopeak, Jinan, China) at laboratorial conditions. The number of seeds germinated were counted every 12 h and continued to 96 h. The seeds were considered germinated when their radicle emerged from seed coat. Finally, the potential of Si-NM-mediated seed germination under acid treatment or control condition was assessed in terms of percentage of seed germination. Root–shoot growth and biomass accumulation of seedlings of two mustard cultivars, BARI mustard-14 and BARI mustard-17, were calculated at 7 days of seed germination.

### 4.2. Description of Field Experiments

The field experiment was conducted in two different agro-ecological zones (AEZ), AEZ-20 (Eastern Surma Kushiara Flood-plan) and AEZ-22 (Northern and Eastern Pedmont-plain), of Bangladesh due to the variation in soil acidity. The soils of AEZ-20 located at (24°54′30.4″ N 91°47′17.5″ E) are slightly acidic (pH ranges from 6.2–6.3) and considered as field soil. In contrast, the soils of AEZ-22 located at (24°54′32.4″ N 91°56′33.5″ E) are strongly acidic (pH ranges from pH 4.5–4.8) and considered as acid soil (Appendix A). The soil acidity status was measured using a glass electrode (Model: HI-2211, Washington Hwy, Smithfield, VA, USA) and a soil to water ratio of 1:2.5 after collecting the three subsamples from each experimental site at a depth of 15 cm, as previously described [41]. The climate of the selected AEZ is Cwa type, humid subtropical with hot summer and cool winter according to Köppen-Geiger climate classification [42]. In strongly acidic soil, we also applied dolomite limestone at a rate of 1 ton ha^−1^ as a world soil correction practice, in addition to Si-NMs sprayed 15 L ha^−1^ at a rate of 0.5 mmol L^−1^ [33]. We spared Si-NMs three times, at seedling stage, vegetative stage and reproductive stage. The field experimental design was a complete randomized block design with six treatments and three replicates. The area of each treatment plot was 10 m square (2.5 m × 4.0 m) in size.

### 4.3. Measurements of Plant Growth and Yield Parameters

The plant biomass and the root–shoot ratio were measured after harvesting of ten plants of two mustard varieties, BARI mustard-14 and BARI mustard-17, from each experimental site at 65 days of sowing. Just after harvesting, the roots were separated from the shoots, washed and air-died at laboratory conditions and weighed in grams using a digital balance (Model-pH 2601, Ningbo, China). The number of pods for each plant, number of seeds per pod and hundred seed weight were estimated after the harvested period of mustard varieties. The counted pods of ten plants of each treatment were averaged and expressed as numbers of pods plant^−1^. Similarly, ten pods per plant and ten replications of each treatment were shelled to extract the seeds and counted after harvest at 90 days of sowing to average the number of seeds per pod. Finally, the hundred seeds of each treatment with three replicates were weighed using an analytical balance and expressed as grams.

### 4.4. Determination of Chlorophyll Content and Net Photosynthetic Rate

To determine the photosynthetic pigment contents, 100 mg of homogenous leaf sample was grounded by using a mortar and pestle with 10 mL of 80% of acetone. Afterward, the homogenate was filtered and we measured the Chl*a* and Chl*b* spectrophotometrically (Model:T80+, PG Instruments Ltd., Wibtft Leicestershir, UK) at a wavelength of 645 nm and 663 nm for the total chlorophyll determination [43]. The net photosynthetic rates (*P*n) were determined on the third fully expanded leaves from the top of plants using an infrared gas analyzer (IRGA) portable photosynthesis system (Model: Li-CHOR 6400, Lincoln, NE, USA) and maintained the conditions, as previously described [4].

### 4.5. Histochemical Staining of H_2_O_2_ and O_2_^•−^ and Measurements of H_2_O_2_, Lipid Peroxidation and Electrolyte Leakage in Mustard Seedlings

The H_2_O_2_ and O_2_^•−^ accumulation in mustard seedlings was visualized by staining with 3, 3-diaminobenzidine (DAB) and nitroblue tetrazolium (NBT) staining, respectively. The H_2_O_2_ seedlings of both mustard varieties were submerged in 1 mg mL^−1^ solution (pH-3.8) after 6 hrs light incubation at 25 °C. In contrast, O_2_^•−^ visualized in seedling by incubating in 0.5 mg mL^−1^ (pH-7.8) solution of NBT after dark adaptation [4]. Quantitative measurements of H_2_O_2_ 0.3 gm of mustard seedlings were homogenized in 3 mL of ice-cooled 1.0 M HClO_4_ solution and centrifuged at 6000× *g* for 5 min in refrigerated conditions (Model: HERMLE Z 326 K, Wehingen, Germany). Afterward, the pH of supernatant was adjusted at 6.5 with KOH and absorbed with 0.05 gm charcoal. Then, the samples were vortexed and centrifuged at 12,000× *g* for 5 min and passed through an AG1.8 pre-packed column. The collected samples were mixed with reaction buffer containing 1 mmol 2,2-azino-di(3-ethylbenzthiazoline-6-sulfonic acid) and 100 mmol potassium acetate (pH-4.4) and 0.25 units of horseradish peroxide at a ration of 1:1 and measured at OD_412_, as previously discussed [4].

To determine the lipid peroxidation, 0.5 g samples were homogenized with 5.0 mL 10% trichloroacetic acid (TCA) and we quantified the MDA equivalents using 2-thiobarbituric acid (TBA), as previously described [44]. After centrifugation at 3000× *g* for 10 min, 4 mL of 20% TCA with 0.65% TBA was added to the 1 mL of the collected supernatant, incubated at 95 °C for 25 min, and we stopped the reaction by placing in an ice bath. We then centrifuged and measured absorbance at 440, 532 and 600 nm. To determine the electrolyte leakage (EL), fresh mustard seedlings were cut into small pieces and placed in a tube with 10 mL of distilled water, with one set shacked at room temperature for 2 h and another set heated at 120 °C for 20 min. We then measured the conductivity EC_1_ and EC_2_ using an EC meter (Model: DDS307, Changsha, China) and calculated as percentage [45].

### 4.6. Antioxidant Assay

The antioxidant activity of extracted samples was measured by 2,2-diphenyl-1-picryl-hydrazyl (DPPH) radical scavenge assay [46]. In brief, 3 mL of 6 × 10^−5^ methanolic acid solution of DPPH was added to the 2 mL of sample extract and left for 30 min in dark at room temperature, and we measured the OD_517_ as sample A. In contrast, the solution without the extract measured the OD as blank and the ability of scavenge DPPH radicals expressed as percentage of total antioxidant activity by following an equation of antioxidant activity (%) = OD_blank_ − OD_sample_/OD_blank_ × 100. The total glutathione (GSH) and oxidized glutathione (GSSG) were measured by using 5,5′-dithio-bis (2-nitrobenzoicacid)-GSSG reductase recycling method and we calculated GSH by deducting GSSG from the total glutathione. The ascorbate (AsA) and dehydroascorbate (DHA) content were also measured spectrophotometrically by following previously described method based on the reduction of Fe^3+^ to Fe^2+^ by AsA in acidic solution [31].

### 4.7. Measurement of Oil Percentage of Mustard Seeds

For the determination of oil percentage, two varieties of mustard seeds were initially cleaned by passing the sieve to remove the sand and stones and air/oven-dried at 60 °C for 10 min. Afterward, seeds were grounded to powder, and the oils of seed flours were extracted with n-hexene in a Soxhlet apparatus for 6 h following the operating conditions specified in IUPAC [47]. The oil was recovered by evaporating the solvent n-hexene by using a rotary evaporator and the extracted oil percentage assay following the equation w_1_ − w_2_/w_1_ × 100, where w_1_ is the initial sample weight placed in the thimble and w_2_ is sample weight after evaporation.

### 4.8. Statistical Analysis

Statistical analyses were performed using SPSS (version 16.0 Inc., Chicago, IL, USA). The normality and homogeneity of data of all variables were checked by using Shaprio–Wilk test. The field experimental design was a completely randomized block design with three replications. The treatment means calculated by averaging the values of three replications and Tukey’s test (*p* < 0.05) were performed to evaluate the treatment significance. All the experiments were replicated twice to validate the results. Different letters were used to denote the significant differences among the treatment means.

## 5. Conclusions

Acidic soils dominantly functioned as a key stress factor, often influencing the soil physical chemical and biological processes, leading to the generation of ROS in plants and, thereby, reducing the crop growth and productivity. However, the utilization of engineered nanomaterials like silicon nanomaterials (Si-NMs) has been hailed as a blessing of modern agricultural intensification by overcoming the ecological barrier. Here, we found that the seed pretreatment with Si-NMs (at 0.5 mmol L^−1^ doses) plays a critical role in redox homeostasis under acidic conditions by stimulating the antioxidant activity and upholding the ratio of GSH/GSSG and AsA/DHA, thus increasing the mustard seed germination, seedling growth and biomass accumulation. Moreover, the exogenous application of selected doses of Si-NMs at the field level showed improved plant growth and biomass accumulation in low soil pH conditions that might be associated with increased chlorophyll content and net photosynthesis. The results also showed that exogenous Si-NMs improves all the critical attributes of crop productivity and economic resilience, such as the total numbers of siliqua per plants, number of seeds per siliqua and hundred seeds weight, seed yield and oil content percentage under stressed conditions in both tested varieties, BARI mustard-14 and BARI mustard-17. These caused further augmentation in plants grown in normal or slightly acidic field conditions. The results also revealed that Si-NMs failed to completely compensate for the yield and yield attributes of mustard crops at extremely low soil pH (pH 4.5–4.8) conditions than the crops grown in field soil (slightly acidic pH 6.2–6.3) conditions. However, dolomite supplementation as a global acid soil correction practice upon Si-NM application not only reduced the crop sensitivity to extremely low soil pH but also improved all the metrics of mustard yields associated with farming resilience (Figure 8). These studies highlighted the potential of Si-NMs to sustainably expand the agricultural intensification of crops in areas affected by low soil pH, essential for global food security and agricultural resilience [37]. However, further study is essential to unveil the cardinal molecular basis salted by Si-NMs before reaching the farm-gate.

## Figures and Tables

**Figure 1 ijms-25-10318-f001:**
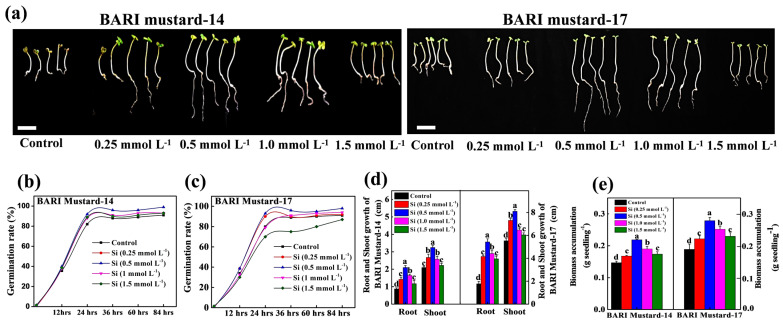
Selection of Si dose based on seed germination and seedling growth. (**a**–**c**) Effect of different doses (0–1.5 mmol L^−1^) of Si nanomaterials (Si-NMs) on seed germination, bars = 1 cm and (**d**,**e**) root shoot growth of seedling of two mustard varieties, BARI-14 and BARI-17, at 7 days of germination. The data presented as the means of three replicates (±SE) and the means denoted same letter do not differ significantly at *p* < 0.05 according to Tukey’s test.

**Figure 2 ijms-25-10318-f002:**
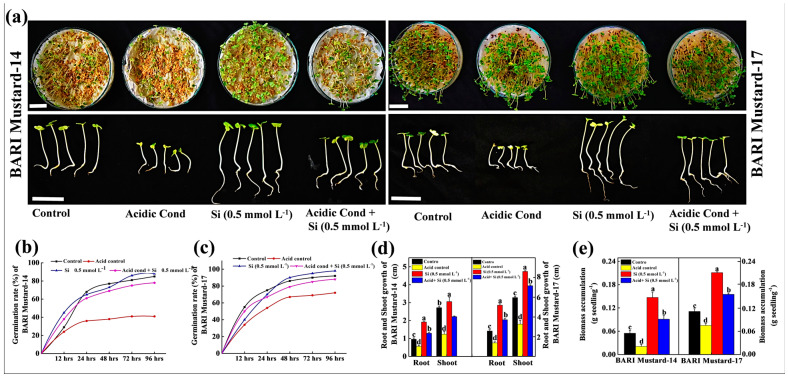
Effect of selected dose (0.5 mmol L^−1^) of Si-NMs on germination and seedling growth under acidic conditions (at pH 4.5); (**a**–**c**) effect of Si-NMs at 0.5 mmol L^−1^ doses on seed germination, bars = 2 cm upper panel and bars = 1 cm lower panel and (**d**,**e**) root shoot growth and biomass accumulation of seedlings of two mustard cultivars, BARI-14 and BARI-17, at 7 days of germination under acidic conditions. The data presented as the means of three replicates (±SE) and the means denoted by the same letter do not differ significantly at *p* < 0.05 according to Tukey’s test.

**Figure 3 ijms-25-10318-f003:**
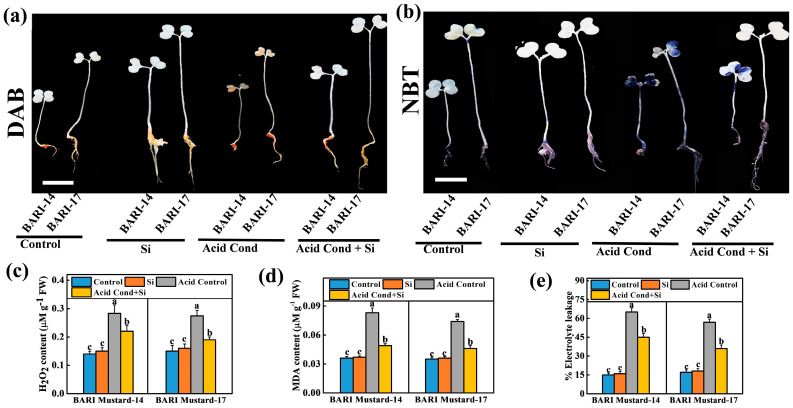
Si-NMs at selected dose (0.5 mmol L^−1^) reduce acid stress (at pH 4.5) induced over accumulation of ROS in mustard seedlings; (**a**,**b**) histochemical staining of mustard seedlings with DAB and NBT, respectively, bars = 1 cm; (**c**) H_2_O_2_ accumulation, (**d**) lipid peroxidation (MDA) content and (**e**) electrolyte leakage (EL) of two mustard varieties, BARI-14 and BARI-17, at 7 days of germination under acidic conditions. The data presented as the means of three replicates (±SE) and the means denoted by the same letter do not differ significantly at *p* < 0.05 according to Tukey’s test.

**Figure 4 ijms-25-10318-f004:**
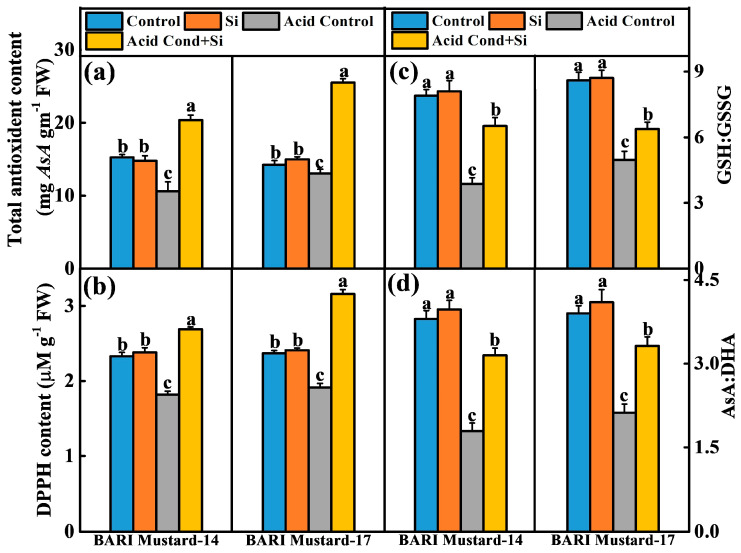
Si-NMs at selected dose (0.5 mmol L^−1^) enhance antioxidant acitivity and cellular redox state in mustard seedlings under acid stress conditions; (**a**) total antoxidant activity, (**b**) DPPH content, (**c**) GSH/GSSG ratio and (**d**) AsA:DHA rato of two mustard varieties, BARI-14 and BARI-17, at 7 days of germination under acidic conditions. The data presented as the means of three replicates (±SE) and the means denoted by the same letter do not differ significantly at *p* < 0.05 according to Tukey’s test.

**Figure 5 ijms-25-10318-f005:**
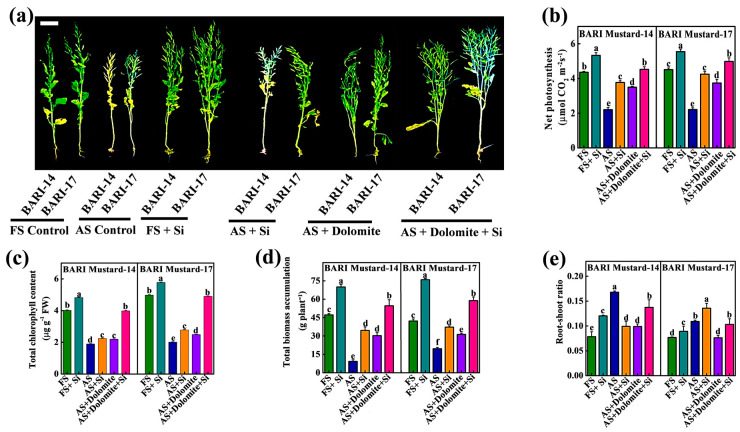
Effect of exogenous application of selected dose (0.5 mmol L^−1^) of Si-NMs on photosynthetic activity and biomass accumulation under field soil (pH 6.2–6.3) and strongly acidic soil (pH 4.5–4.8) conditions; (**a**) photographic image of mustard plants, bar = 24 cm; (**b**) net photosynthetic rate, (**c**) total chlorophyll content, (**d**) biomass accumulation and (**e**) root–shoot ratio of two mustard varieties, BARI-14 and BARI-17, at 60 days of sowing. The data presented as the means of three replicates (±SE) and the means denoted by the same letter do not differ significantly at *p* < 0.05 according to Tukey’s test.

**Figure 6 ijms-25-10318-f006:**
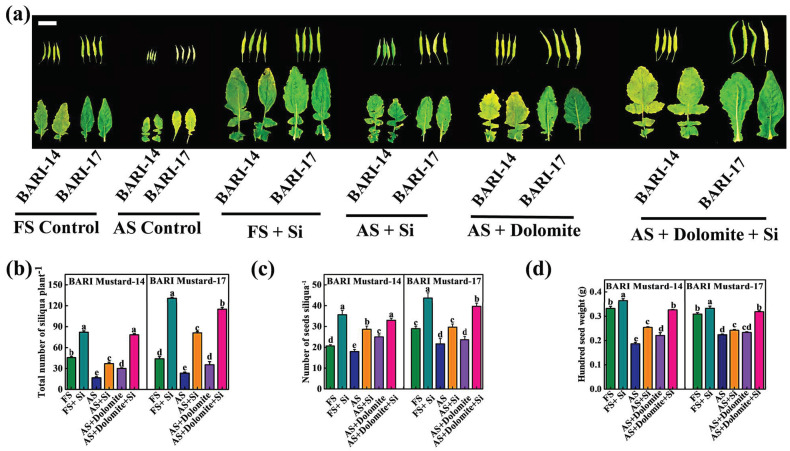
Exogenous Si-NMs at 0.5 mmol L^−1^ dose enhanced yield attribute characteristics of mustard crop under acidic field soil conditions. (**a**) Photograph of siliqua and leaves of mustard crops, bar = 2 cm, (**b**) total number of siliqua per plants, (**c**) numbers of seeds per siliqua and (**d**) hundred seeds weight after harvest of two mustard varieties, BARI-14 and BARI-17, cultivated at two different field soil conditions, having soil pH 4.5–4.8 and pH 6.2–6.3, respectively. The data presented as the means of three replicates (±SE) and the means denoted by the same letter do not differ significantly at *p* < 0.05 according to Tukey’s test.

**Figure 7 ijms-25-10318-f007:**
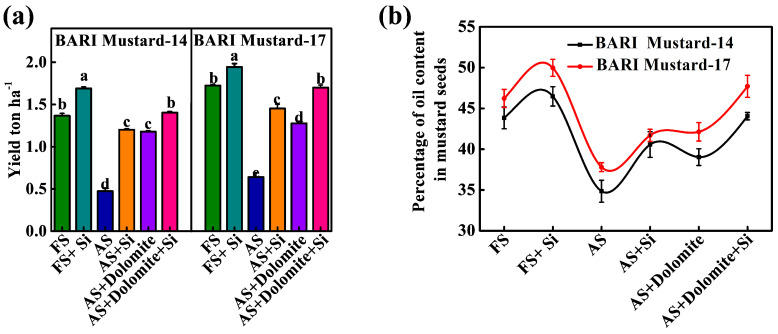
Si-NMs promote seed yield and oil content restricted by soil acidity; (**a**) total seed yield and (**b**) percent of oil content of seeds of two mustard varieties, BARI-14 and BARI-17, cultivated at two different field soil conditions, having soil pH 4.5–4.8 and pH 6.2–6.3, respectively. The data presented as the means of three replicates (±SE) and the means denoted by the same letter do not differ significantly at *p* < 0.05 according to Tukey’s test.

**Figure 8 ijms-25-10318-f008:**
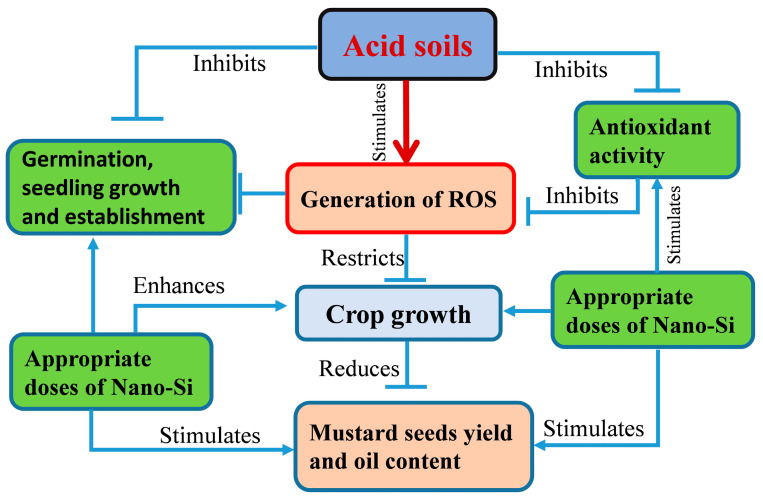
A proposed model depicting the involvement of Si-NMs in reduction in crop sensitivity to extremely low soil pH. Exogenous Si-NM application at 0.5 mmol L^−1^ doses enhanced ROS homeostasis by promoting enzymatic activity and thereby improves all the metrics of mustard yields associated with farming resilience.

## Data Availability

Data is contained within the article and Appendix A.

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
