# Peer review of "Silicon Nanomaterials Enhance Seedling Growth and Plant Adaptation to Acidic Soil by Promoting Photosynthesis and Antioxidant Activity in Mustard (Brassica campestris L.)"

_ijms, 2024, doi:10.3390/ijms251910318_

Round 1

Reviewer 1 Report

Comments and Suggestions for Authors

This work assessed the growth-promoting effect of Si-NM on mustard in the acid soil. The article is informative but has some problems to be solved. Comments are listed below.

1) There are too many typo and grammar errors in the manuscript. Revise the manuscript carefully.

2) It is customary to use the unit “mM” for mmol/L in articles. The unit “mM/L” or “mM L-1” in this manuscript is rarely seen elsewhere.

3) The method for field experiments should be revised. What does the description “Si-NMs spray at a rate of 0.5 mM L-1” mean? How many volume of Si-NM was used per hectare? Was Si-NM sprayed once or repeatedly?

4) Give the bar length in Fig1 & 2.

5) Provide the meaning of “adaptive response”. Was it good or bad for the health of mustard? Besides, the plot of Fig 8 should be revised. “Mustard seeds yield and oil content” is inhibited by “Adaptive response” that is inhibited by “Generation of ROS”. The roles of “Adaptive response” in Fig8 seem contradictory.

6) Changes in pH and other properties of the acid soil should be provided. The content of Si in different tissue of mustard should be determined at different growth stages.

Comments on the Quality of English Language

There are too many typo and grammar errors in the manuscript.

Author Response

Dear Reviewer

Thank you for critical review and suggestions of our submitted MS. All the suggestions/comments have been inculcated and addressed carefully. 

The point by point response are as follows

Comments and Suggestions for Authors

This work assessed the growth-promoting effect of Si-NM on mustard in the acid soil. The article is informative but has some problems to be solved. Comments are listed below.

Ans: Thank you for your critical review and finds the article informative. All the suggestions/comments have been inculcated and addressed carefully. 

1) There are too many typo and grammar errors in the manuscript. Revise the manuscript carefully.

Ans: Thank you very much much for the concern, in revised version of the MS we have checked and corrected the typo and grammar errors

2) It is customary to use the unit “mM” for mmol/L in articles. The unit “mM/L” or “mM L-1” in this manuscript is rarely seen elsewhere.

Ans: Thank you in revised version of the MS we have customized the unit and write mmol L-1 instead of mM L-1.

3) The method for field experiments should be revised. What does the description “Si-NMs spray at a rate of 0.5 mM L-1” mean? How many volume of Si-NM was used per hectare? Was Si-NM sprayed once or repeatedly?

Ans: Thank you for the concern. In revised version of the MS we have added the volume (15 L) that we have sprayed each time per hector and total three times (seedling stage, vegetative stage and reproductive stage) sprayed at a rate of 0.5 mmol L-1.

4) Give the bar length in Fig1 & 2.

Ans: Thank you, in the revised version of the MS we have added the bar length in legends of figure 1 and figure 2.

5) Provide the meaning of “adaptive response”. Was it good or bad for the health of mustard? Besides, the plot of Fig 8 should be revised. “Mustard seeds yield and oil content” is inhibited by “Adaptive response” that is inhibited by “Generation of ROS”. The roles of “Adaptive response” in Fig8 seem contradictory.

Ans: We are apologies for the mistake, Adaptive response is obviously good for successful crop production, in the revised version of the MS we have corrected the figure 8 by replacing the ‘Adaptive response’ by the term ‘Crop growth’. 

6) Changes in pH and other properties of the acid soil should be provided. The content of Si in different tissue of mustard should be determined at different growth stages.

Ans: Thank you for the comments, in revised version of MS we have added the soil properties as a supplementary file. And the Si founds nontoxic and has no health hazard issues (doi.10.1016/j.envres.2023.116292) hence, in present study we do not measure the Si content in various growth stage of plants. However, we have plan to do the independent experiment in near future, where we must focused on the aforementioned issues. 

Comments on the Quality of English Language

There are too many typo and grammar errors in the manuscript.

Ans: Thank you, in revised version of the MS we have carefully checked and corrected the typo and grammar errors.

I shall be highly thankful for considering the revised MS as suitable for publication in the journal ‘IJMS’.

Thank you.

Yours sincerely,

Md. Kamrul Hasan, Ph.D

Sylhet Agricultural University

Reviewer 2 Report

Comments and Suggestions for Authors

Manuscript ID ijms-3199455 entitled 'Silicon nanomaterials enhance seedling growth and plant adaptation to acidic soil by promoting photosynthesis and antioxidant activity in mustard (Brassica Campestris L.)' is a good scientific article. It falls within the scope of the special issue. It deserves to be published in IJMS. My comments on the individual chapters are as follows.

The abstract needs some improvement. The abbreviations in it should be explained the first time they are used.

The introduction does a good job of introducing the reader to the issues presented in the manuscript. It needs no additions.

Materials and Methods. A weakness of the experiment is that it was carried out on soils in two different ecological zones. It would have been better to compare the effects of the tested factors in soils of one ecological zone with different pH. A more detailed characterisation of the soils used in the field experiment is lacking. This would have made it clear whether the attributed changes were the exclusive domain of acidity or whether they were also the result of the interaction of other factors. The size of the study plots is also not described. The characteristics of the Si-NM should be presented in this chapter. The unit of mass 'g' should be used instead of 'gm'.

Results. The chapter is well written and illustrated. However, the quality of figures 2b-e and 6b-d should be improved.

Discussion. The chapter has interpreted the results of its own research well and compared them with the results of other researchers.

Conclusions. Conclusions should only be drawn from the experiment described in the manuscript, therefore no literature should be used in this chapter.

References. The reference list should be prepared according to the requirements of the IJMS. In the current version it is presented in different ways.

Author Response

Dear Reviewer

Thank you for critical review and suggestions of our submitted MS. All the suggestions/comments have been inculcated and addressed carefully. 

Comments and Suggestions for Authors

Manuscript ID ijms-3199455 entitled 'Silicon nanomaterials enhance seedling growth and plant adaptation to acidic soil by promoting photosynthesis and antioxidant activity in mustard (Brassica Campestris L.)' is a good scientific article. It falls within the scope of the special issue. It deserves to be published in IJMS. My comments on the individual chapters are as follows.

Ans: Thank you for the critical review and find the article suitable within the scope of the special issue. However, all the suggestions/comments have been inculcated and addressed carefully.

The abstract needs some improvement. The abbreviations in it should be explained the first time they are used.

Ans: Thank you for the concern, in revised version of the MS we have explained the abbreviations at first time of their used.

The introduction does a good job of introducing the reader to the issues presented in the manuscript. It needs no additions.

Ans: Thank you that the reviewer finds the introduction good and need no additions.

Materials and Methods. A weakness of the experiment is that it was carried out on soils in two different ecological zones. It would have been better to compare the effects of the tested factors in soils of one ecological zone with different pH. A more detailed characterisation of the soils used in the field experiment is lacking. This would have made it clear whether the attributed changes were the exclusive domain of acidity or whether they were also the result of the interaction of other factors. The size of the study plots is also not described. The characteristics of the Si-NM should be presented in this chapter. The unit of mass 'g' should be used instead of 'gm'.

Ans: Thank you for the concern, although, we have conducted the experiment in two different agro ecological zones the distance between the zones is only 10 kilometer and there is no mentionable weather changes. However, in revised version of the MS we added the soil properties as a supplementary file for more details. Moreover we have also mention the plot size in revised version of the MS and the unit of mass 'g' used instead of 'gm'.

Results. The chapter is well written and illustrated. However, the quality of figures 2b-e and 6b-d should be improved.

Ans: Thank you that the reviewer finds the results well written and illustrated. We have improved the all the graph and write unit of mass 'g' used instead of 'gm' including figures 2b-e and 6b-d.

Discussion. The chapter has interpreted the results of its own research well and compared them with the results of other researchers.

Ans: Thank you very much for the comments that the reviewer finds the discussion chapter well interpretation and comparison with the results of other researchers.

Conclusions. Conclusions should only be drawn from the experiment described in the manuscript, therefore no literature should be used in this chapter.

Ans: Thank you for the comments, we have deleted the reference from conclusion in revised version of the MS.

References. The reference list should be prepared according to the requirements of the IJMS. In the current version it is presented in different ways.

Ans: Thank you, in revised version of the MS we prepare the reference according to the requirements of the IJMS.

I shall be highly thankful for considering the revised MS as suitable for publication in the journal ‘IJMS’.

Thank you.

Yours sincerely,

Md. Kamrul Hasan, Ph.D

Sylhet Agricultural University

Reviewer 3 Report

Comments and Suggestions for Authors

The manuscript is prepared on a very interesting and current topic. It can be beneficial to the field of study. However, before accepting it, I have the following comments and suggestions for editing the manuscript:  

Introduction – in the second paragraph, an important source is /7/ Chakraborty et al. (2024), but there are also studies devoted to e.g. aluminum on root growth in sunflower (DOI: 10.17221/110/2021-CJGPB), etc., which the authors could use, and not just take information from the "review" type article for part line 57-62. In the next paragraph, the authors focus on silicone materials, but there are more types of nanomaterials. Even with regard to the hypothesis that the authors formulate in the section (line 76-79), inspiration could also be found in other nanoparticles, e.g. graphene oxide (GO) and a study in buckwheat, where the authors described an effect on genes related to ROS (DOI: 10.17221/ 69/2024-CJGPB), i.e. that it would be possible to formulate whether there is a similar effect in the case of silicon nanomaterials.  

Materials and Methods – subsection 2.2 states that a glass electrode measurement was performed, but the type and manufacturer of the pH meter is crucial, which is missing. In section 2.3, it would be appropriate to indicate the standardized growth phases /internationally recognized scale/ when the given parameters were evaluated. In subsections 2.4 to 2.7, the number of repetitions and technical replicates is missing, which is a very important figure. A certain number of repetitions is indicated only in the graphic appendices in the results.  

Results - they are well described, but the very frequent inclusion of links to graphical appendices, which are only later in the text, is problematic. Please always place images after the appropriate link in the text. Figure 1-2 (b-e), Figure 3 (c-e), Figure 5 (c-e) and Figure 6 (b-d) could be larger. Why is the variability not indicated in Figure 1 (b-c) and Figure (b-c), similar to what the authors have indicated in Figure 7b? After all, the assessment had repetitions and there was certainly some variability in the values, right?  

The discussion is adequate and I don't think it's even necessary to refer to all Figures again. Perhaps only in the section (line 490-492) the authors describe the effect of pH on root growth and microbial activity, so they could also mention the information that this fact also plays a significant role for the mobility of PTEs (potentially toxic elements), as they demonstrated for cannabis in the study from this year (DOI: 10.1196/s40538-024-00544-6). The area of ​​phytoremediation is gaining great importance due to soil pollution/contamination.  

Conclusion - it's OK, but again Figure 8 is wrongly included.   References - in my opinion, they are not processed according to the guidelines for authors. It is not possible to list only the first author and "et al." No, standards are always respected in writing journal titles, e.g. Ref. 14, etc. this part needs careful editing.  

Formal comments:

line 4 - Brassica Campestris - properly campestris

line 18 - siNMs - abbreviation used for the first time, but explained only in the Introduction section. All abbreviations should be explained the first time they are used, and this also applies to the abstract. Then it is not necessary to mention it in the Introduction section.

line 43 - ROS - abbreviation not explained. Please check the manuscript carefully from a formal point of view.

Comments on the Quality of English Language

I also recommend careful proofreading of the English language to remove typos and respect the standards that Latin names are written in italics, e.g. in vitro, etc.

Author Response

Dear Reviewer

Thank you for critical review and suggestions of our submitted MS. All the suggestions/comments have been inculcated and addressed carefully. 

Comments and Suggestions for Authors

The manuscript is prepared on a very interesting and current topic. It can be beneficial to the field of study. However, before accepting it, I have the following comments and suggestions for editing the manuscript:  

Thank you for critical review and valuable suggestions of our submitted MS. We are pleased to hear that the reviewer finds our study interesting and would be beneficial to the field study. However, all the suggestions/comments have been inculcated and addressed carefully. 

Introduction – in the second paragraph, an important source is /7/ Chakraborty et al. (2024), but there are also studies devoted to e.g. aluminum on root growth in sunflower (DOI: 10.17221/110/2021-CJGPB), etc., which the authors could use, and not just take information from the "review" type article for part line 57-62. In the next paragraph, the authors focus on silicone materials, but there are more types of nanomaterials. Even with regard to the hypothesis that the authors formulate in the section (line 76-79), inspiration could also be found in other nanoparticles, e.g. graphene oxide (GO) and a study in buckwheat, where the authors described an effect on genes related to ROS (DOI: 10.17221/ 69/2024-CJGPB), i.e. that it would be possible to formulate whether there is a similar effect in the case of silicon nanomaterials.  

Ans: Thank you for the concern, In introduction we have added the reference (DOI: 10.17221/110/2021-CJGPB) at 9, that focused on the root growth in sunflower under aluminum stress.  However, we focused mainly the silicon-nanomaterial’s hence during the draw of hypothesis of the experiment we avoid other nanomaterial’s many of which has also a stimulatory effect on crop growth under various stress situation.

Materials and Methods – subsection 2.2 states that a glass electrode measurement was performed, but the type and manufacturer of the pH meter is crucial, which is missing. In section 2.3, it would be appropriate to indicate the standardized growth phases /internationally recognized scale/ when the given parameters were evaluated. In subsections 2.4 to 2.7, the number of repetitions and technical replicates is missing, which is a very important figure. A certain number of repetitions is indicated only in the graphic appendices in the results.  

Ans: Thank you for the concern, in revised version of the MS we have added the model number of glass electrode ((Model: HI-2211, Rhode Island, USA). In revised version of the MS we have indicated the the standardized growth phases /internationally recognized scale in section 2.3. In statistical  analytical section we hacve clearly mention that the treatment means calculated by averaging the values of three replications, so we do not add the same information in sub section 2.4 to 2.7 for avoiding repetition.

Results - they are well described, but the very frequent inclusion of links to graphical appendices, which are only later in the text, is problematic. Please always place images after the appropriate link in the text. Figure 1-2 (b-e), Figure 3 (c-e), Figure 5 (c-e) and Figure 6 (b-d) could be larger. Why is the variability not indicated in Figure 1 (b-c) and Figure (b-c), similar to what the authors have indicated in Figure 7b? After all, the assessment had repetitions and there was certainly some variability in the values, right?  

Ans: Thank you very much that the reviewer finds the results well described. However the frequently inclusion of links to graphical appendices due to the easy understanding of the general reader. In revised version of MS we have tried to shortage such a frequent inclusion of links to graphical appendices.

The discussion is adequate and I don't think it's even necessary to refer to all Figures again. Perhaps only in the section (line 490-492) the authors describe the effect of pH on root growth and microbial activity, so they could also mention the information that this fact also plays a significant role for the mobility of PTEs (potentially toxic elements), as they demonstrated for cannabis in the study from this year (DOI: 10.1196/s40538-024-00544-6). The area of ​​phytoremediation is gaining great importance due to soil pollution/contamination.  

Ans: Thank you for your nice comments, in revised version of the MS we have added the reference (https://doi.org/10.1186/s40538-024-00544-6) in as suggested by the reviewer in number 42. 

Conclusion - it's OK, but again Figure 8 is wrongly included.   References - in my opinion, they are not processed according to the guidelines for authors. It is not possible to list only the first author and "et al." No, standards are always respected in writing journal titles, e.g. Ref. 14, etc. this part needs careful editing.  

Ans: We are apologies for the mistake, Adaptive response is obviously good for successful crop production, In the revised version of the MS we have corrected the figure 8 by replacing the ‘Adaptive response’ by the term ‘Crop growth’.  And in revised version of the MS we prepare the reference according to the requirements of the IJMS.

Formal comments:

line 4 - Brassica Campestris - properly campestris

Ans: Thank you for the concern, we have properly write the scientific name Brassica campestris.

line 18 - siNMs - abbreviation used for the first time, but explained only in the Introduction section. All abbreviations should be explained the first time they are used, and this also applies to the abstract. Then it is not necessary to mention it in the Introduction section.

Ans: Thank you for the concern, in revised version of the MS we have explained the abbreviations at first time of their used.

line 43 - ROS - abbreviation not explained. Please check the manuscript carefully from a formal point of view.

Ans: Thank you for the concern, in revised version of the MS we have explained the abbreviations at first time of their used.

Comments on the Quality of English Language

I also recommend careful proofreading of the E

Ans: Thank you, in revised version of the MS we have carefully checked and corrected the typo and grammar errors if there is any.

I shall be highly thankful for considering the revised MS as suitable for publication in the journal ‘IJMS’.

Thank you.

Yours sincerely,

Md. Kamrul Hasan, Ph.D

Sylhet Agricultural University

Round 2

Reviewer 1 Report

Comments and Suggestions for Authors

The authors submitted the revised manuscript hastily. Some questions are still unresolved.

1) Since the acid soil affected the root more directly than the aboveground tissue, the spraying method became important. The authors sprayed Si-NM three times in the field experiment. It is unclear where they sprayed. On the aboveground tissue of mustard or on the topsoil did they spray? If Si-NM was sprayed on the stem and leaves, did Si-NM affect the antioxidant activity of root? Otherwise, if Si-NM was sprayed on the topsoil, did the pH and physiochemical properties of soil vary after spraying? Besides, amendment of the soil by dolomite undoubtly changed the property of soil. All the missing data should be provided.

2) The meaning of Fig8 is unclear. i) Does “Crop growth” reduce “Mustard seed yield and oil content”? In the flowchart, double inhibition means activation. Hence, acid soil enhances the yield of mustard seeds by generating ROS according to Fig8. ii) Acid soil inhibits seed germination and growth, and inhibits the antioxidant activity as well. Is the inhibition of germination and growth due to low level of antioxidant activity? If it is, the flowchart is unreasonable.

3) Grammar errors still exist after revision. Please check them again.

Comments on the Quality of English Language

Grammar errors still exist after revision. Please check them again.

Author Response

Dear Reviewer

We are apologize for the mistake and thank you again for critical review and suggestions of our submitted MS. All the suggestions/comments have been inculcated and addressed carefully. 

The point by point response are as follows

Comments and Suggestions for Authors

This work assessed the growth-promoting effect of Si-NM on mustard in the acid soil. The article is informative but has some problems to be solved. Comments are listed below.

Ans: Thank you for your critical review and finds the article informative. All the suggestions/comments have been inculcated and addressed carefully. 

1) There are too many typo and grammar errors in the manuscript. Revise the manuscript carefully.

Ans: Thank you very much much for the concern, in revised version of the MS we have checked and corrected the typo and grammar errors

2) It is customary to use the unit “mM” for mmol/L in articles. The unit “mM/L” or “mM L-1” in this manuscript is rarely seen elsewhere.

Ans: Thank you in revised version of the MS we have customized the unit and write mmol L-1 instead of mM L-1.

3) The method for field experiments should be revised. What does the description “Si-NMs spray at a rate of 0.5 mM L-1” mean? How many volume of Si-NM was used per hectare? Was Si-NM sprayed once or repeatedly?

Ans: Thank you for the concern. In revised version of the MS we have added the volume (15 L) that we have sprayed each time per hector and total three times (seedling stage, vegetative stage and reproductive stage) sprayed at a rate of 0.5 mmol L-1.

4) Give the bar length in Fig1 & 2.

Ans: Thank you, in the revised version of the MS we have added the bar length in legends of figure 1 and figure 2.

5) Provide the meaning of “adaptive response”. Was it good or bad for the health of mustard? Besides, the plot of Fig 8 should be revised. “Mustard seeds yield and oil content” is inhibited by “Adaptive response” that is inhibited by “Generation of ROS”. The roles of “Adaptive response” in Fig8 seem contradictory.

Ans: We are apologies for the mistake, Adaptive response is obviously good for successful crop production, in the revised version of the MS we have corrected the figure 8 by replacing the ‘Adaptive response’ by the term ‘Crop growth’. 

6) Changes in pH and other properties of the acid soil should be provided. The content of Si in different tissue of mustard should be determined at different growth stages.

Ans: Thank you for the comments, in revised version of MS we have added the soil properties as a supplementary file. And the Si founds nontoxic and has no health hazard issues (doi.10.1016/j.envres.2023.116292) hence, in present study we do not measure the Si content in various growth stage of plants. However, we have plan to do the independent experiment in near future, where we must focused on the aforementioned issues. 

Comments on the Quality of English Language

There are too many typo and grammar errors in the manuscript.

Ans: Thank you, in revised version of the MS we have carefully checked and corrected the typo and grammar errors.

I shall be highly thankful for considering the revised MS as suitable for publication in the journal ‘IJMS’.

Thank you.

Yours sincerely,

Md. Kamrul Hasan, Ph.D

Sylhet Agricultural University

Reviewer 3 Report

Comments and Suggestions for Authors

The authors accepted most of my comments, or adequately justified their decision. Nevertheless, I would once again point out some essential things (especially of a formal nature):

Please always move the figure in the text after the first link to the given figure. It is not standard for the text to have figures first and then only the reference in the subsequent text. This is a formal matter, but from the point of view of clarity for the reader of the manuscript (article), it is essential.

The comment on Figure 1b,c, respectively, was not accepted or explained. Figure 2b,c - where there is no indication of parameter variability as in Figure 7b.

The References section has been significantly improved, i.e. additions by all authors of the article, but it is still necessary to make some corrections and additions, e.g. Ref. 12, 32, etc. - the correct name of the journal is not given (i.e. writing upper and lower case letters in the title); Ref. 9 - is incomplete. The authors, title and year of the article are listed, but the journal, year and pages are missing. The Refrences section should be checked carefully.

I know that my comments are mainly of a formal nature, but I still recommend the manuscript to be published after major revision.

Author Response

Dear Reviewer

We are apologize for the mistake and thank you again for critical review and suggestions of our submitted MS

Comments and Suggestions for Authors

The manuscript is prepared on a very interesting and current topic. It can be beneficial to the field of study. However, before accepting it, I have the following comments and suggestions for editing the manuscript:  

Thank you for critical review and valuable suggestions of our submitted MS. We are pleased to hear that the reviewer finds our study interesting and would be beneficial to the field study. However, all the suggestions/comments have been inculcated and addressed carefully. 

Introduction – in the second paragraph, an important source is /7/ Chakraborty et al. (2024), but there are also studies devoted to e.g. aluminum on root growth in sunflower (DOI: 10.17221/110/2021-CJGPB), etc., which the authors could use, and not just take information from the "review" type article for part line 57-62. In the next paragraph, the authors focus on silicone materials, but there are more types of nanomaterials. Even with regard to the hypothesis that the authors formulate in the section (line 76-79), inspiration could also be found in other nanoparticles, e.g. graphene oxide (GO) and a study in buckwheat, where the authors described an effect on genes related to ROS (DOI: 10.17221/ 69/2024-CJGPB), i.e. that it would be possible to formulate whether there is a similar effect in the case of silicon nanomaterials.  

Ans: Thank you for the concern, In introduction we have added the reference (DOI: 10.17221/110/2021-CJGPB) at 9, that focused on the root growth in sunflower under aluminum stress.  However, we focused mainly the silicon-nanomaterial’s hence during the draw of hypothesis of the experiment we avoid other nanomaterial’s many of which has also a stimulatory effect on crop growth under various stress situation.

Materials and Methods – subsection 2.2 states that a glass electrode measurement was performed, but the type and manufacturer of the pH meter is crucial, which is missing. In section 2.3, it would be appropriate to indicate the standardized growth phases /internationally recognized scale/ when the given parameters were evaluated. In subsections 2.4 to 2.7, the number of repetitions and technical replicates is missing, which is a very important figure. A certain number of repetitions is indicated only in the graphic appendices in the results.  

Ans: Thank you for the concern, in revised version of the MS we have added the model number of glass electrode ((Model: HI-2211, Rhode Island, USA). In revised version of the MS we have indicated the the standardized growth phases /internationally recognized scale in section 2.3. In statistical  analytical section we hacve clearly mention that the treatment means calculated by averaging the values of three replications, so we do not add the same information in sub section 2.4 to 2.7 for avoiding repetition.

Results - they are well described, but the very frequent inclusion of links to graphical appendices, which are only later in the text, is problematic. Please always place images after the appropriate link in the text. Figure 1-2 (b-e), Figure 3 (c-e), Figure 5 (c-e) and Figure 6 (b-d) could be larger. Why is the variability not indicated in Figure 1 (b-c) and Figure (b-c), similar to what the authors have indicated in Figure 7b? After all, the assessment had repetitions and there was certainly some variability in the values, right?  

Ans: Thank you very much that the reviewer finds the results well described. However the frequently inclusion of links to graphical appendices due to the easy understanding of the general reader. In revised version of MS we have tried to shortage such a frequent inclusion of links to graphical appendices.

The discussion is adequate and I don't think it's even necessary to refer to all Figures again. Perhaps only in the section (line 490-492) the authors describe the effect of pH on root growth and microbial activity, so they could also mention the information that this fact also plays a significant role for the mobility of PTEs (potentially toxic elements), as they demonstrated for cannabis in the study from this year (DOI: 10.1196/s40538-024-00544-6). The area of ​​phytoremediation is gaining great importance due to soil pollution/contamination.  

Ans: Thank you for your nice comments, in revised version of the MS we have added the reference (https://doi.org/10.1186/s40538-024-00544-6) in as suggested by the reviewer in number 42. 

Conclusion - it's OK, but again Figure 8 is wrongly included.   References - in my opinion, they are not processed according to the guidelines for authors. It is not possible to list only the first author and "et al." No, standards are always respected in writing journal titles, e.g. Ref. 14, etc. this part needs careful editing.  

Ans: We are apologies for the mistake, Adaptive response is obviously good for successful crop production, In the revised version of the MS we have corrected the figure 8 by replacing the ‘Adaptive response’ by the term ‘Crop growth’.  And in revised version of the MS we prepare the reference according to the requirements of the IJMS.

Formal comments:

line 4 - Brassica Campestris - properly campestris

Ans: Thank you for the concern, we have properly write the scientific name Brassica campestris.

line 18 - siNMs - abbreviation used for the first time, but explained only in the Introduction section. All abbreviations should be explained the first time they are used, and this also applies to the abstract. Then it is not necessary to mention it in the Introduction section.

Ans: Thank you for the concern, in revised version of the MS we have explained the abbreviations at first time of their used.

line 43 - ROS - abbreviation not explained. Please check the manuscript carefully from a formal point of view.

Ans: Thank you for the concern, in revised version of the MS we have explained the abbreviations at first time of their used.

Comments on the Quality of English Language

I also recommend careful proofreading of the E

Ans: Thank you, in revised version of the MS we have carefully checked and corrected the typo and grammar errors if there is any.

I shall be highly thankful for considering the revised MS as suitable for publication in the journal ‘IJMS’.

Thank you.

Yours sincerely,

Md. Kamrul Hasan, Ph.D

Sylhet Agricultural University

Round 3

Reviewer 1 Report

Comments and Suggestions for Authors

The authors didn’t make response to any of my comments. I don’t know why they submitted a revision without any changes. The only difference of the revision is the paper structure.

Reviewer 3 Report

Comments and Suggestions for Authors

The authors did not respond, or only partially to my notes and comments in Review #2. Instead, they re-inserted my responses and reactions to Review #1. Therefore, I am posting my slightly modified Review #2 and expect it to be adequately addressed by the authors.

-----

The authors accepted most of my comments, or adequately justified their decision. Nevertheless, I would once again point out some essential things (especially of a formal nature):

Please always move the figure in the text after the first link to the given figure. It is not standard for the text to have figures first and then only the reference in the subsequent text (Figure 1 and 8). This is a formal matter, but from the point of view of clarity for the reader of the manuscript (article), it is essential.

The comment on Figure 1b,c, respectively, was not accepted or explained. Figure 2b,c - where there is no indication of parameter variability as in Figure 7b.

The References section has been significantly improved, i.e. additions by all authors of the article, but it is still necessary to make some corrections and additions, e.g. Ref. 12, 32, etc. - the correct name of the journal is not given (i.e. writing upper and lower case letters in the title); Ref. 9 - is incomplete. The authors, title and year of the article are listed, but the journal, year and pages are missing. The References section should be checked carefully.

I know that my comments are mainly of a formal nature, but I still recommend the manuscript to be published after major revision.